# AdaSent: Efficient Domain-Adapted Sentence Embeddings for Few-Shot Classification

**Yongxin Huang[1], Kexin Wang[1], Sourav Dutta[2],**
**Raj Nath Patel[2], Goran Glavaš[3], Iryna Gurevych[1]**

[1]Ubiquitous Knowledge Processing Lab (UKP Lab)
Department of Computer Science and Hessian Center for AI (hessian.AI)
Technical University of Darmstadt
[2]Huawei Research Centre, Dublin, Ireland
[3]Center for AI and Data Science, University of Würzburg
[1]www.ukp.tu-darmstadt.de
[2]{sourav.dutta2,raj.nath.patel}@huawei.com
[3]goran.glavas@uni-wuerzburg.de

## Abstract

Recent work has found that few-shot sentence classification based on pre-trained Sentence Encoders (SEs) is efficient, robust, and effective. In this work, we investigate strategies for domain-specialization in the context of few-shot sentence classification with SEs. We first establish that unsupervised Domain-Adaptive Pre-Training (DAPT) of a base Pre-trained Language Model (PLM) (i.e., not an SE) substantially improves the accuracy of few-shot sentence classification by up to 8.4 points. However, applying DAPT on SEs, on the one hand, disrupts the effects of their (general-domain) Sentence Embedding Pre-Training (SEPT). On the other hand, applying general-domain SEPT on top of a domain-adapted base PLM (i.e., after DAPT) is effective but inefficient, since the computationally expensive SEPT needs to be executed on top of a DAPT-ed PLM of each domain. As a solution, we propose AdaSent, which decouples SEPT from DAPT by training a SEPT adapter on the base PLM. The adapter can be inserted into DAPT-ed PLMs from any domain. We demonstrate AdaSent's effectiveness in extensive experiments on 17 different few-shot sentence classification datasets. AdaSent matches or surpasses the performance of full SEPT on DAPT-ed PLM, while substantially reducing the training costs. The code for AdaSent is available[1].

## 1  Introduction

Few-shot learning aims at training an effective model with a few labeled examples, reducing the cost of developing models for new domains and tasks. In recent work, SetFit (Tunstall et al., 2022) achieves strong performance in few-shot classification by contrastively fine-tuning (Koch et al., 2015)

[1]https://github.com/UKPLab/AdaSent

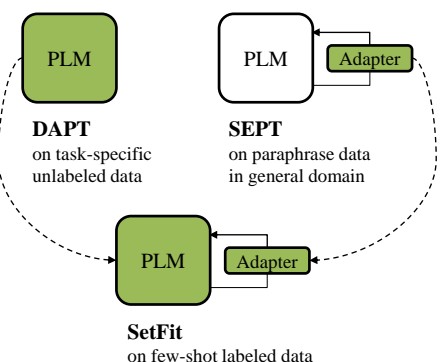

Figure 1: Training diagram of AdaSent. Trainable parameters are marked in green. After Domain-Adaptive Pre-training (DAPT) on the Pre-Trained Language Model (PLM) and Sentence-Embedding Pre-Training (SEPT) with an adapter, the two parts are assembled together to perform SetFit for few-shot classification.

pre-trained sentence embeddings. Being prompt-free and effective on relative small models, Set-Fit is much more efficient than popular prompt-based methods including In-Context Learning (ICL, Brown et al., 2020) and Pattern Exploit Training (PET, Schick and Schütze, 2021), which require careful prompt engineering and large model size.

Despite its success, SetFit fine-tunes a sentence encoder with only a few labeled samples without leveraging unlabeled data from the target-task domain, which are easy to obtain. It is well-known that Domain-Adaptive Pre-Training (DAPT)[2] on a vanilla PLM with unlabeled in-domain data can significantly improve its downstream performance (Han and Eisenstein, 2019; Gururangan et al., 2020). However, it is ineffective to apply

[2]By DAPT we refer to the TAPT (Task-Adaptive Pre-Training) in Gururangan et al. (2020). We do not strictly differentiate between domain and task in the present work.

DAPT on sentence encoders, i.e. vanilla PLMs that have undergone Sentence Embedding Pre-Training (SEPT, Reimers and Gurevych, 2019) in general domain, as DAPT messes up the effects of SEPT and disrupts the model's ability to semantically accurately embed sentences. Though DAPT *before* SEPT is effective in contrast (Wang et al., 2021), it is computationally inefficient as the general-domain SEPT has to be done all over again on every domain-adapted PLM if we have more than one domain.

To create a domain-specialized sentence encoder for few-shot sentence classification both efficiently and effectively, we propose *AdaSent*, which combines DAPT and SEPT in a modular fashion. Specifically, it stores the sentence-specialization abilities – obtained via a single SEPT procedure in the general domain – into an adapter. This sentence-encoding adapter is trained once regardless of the number of domains, and can be plugged into domain-adapted PLMs from various domains to make them domain-specialized sentence encoders, on which SetFit is carried out to do downstream classification training (Figure 1). Our experiments show that AdaSent can match or surpass the inefficient "full SEPT *after* DAPT" approach's performance on 17 sentence classification tasks from various domains. The contribution of AdaSent is two-fold:

- AdaSent significantly improves SetFit, the previous state-of-the-art few-shot classification approach, by leveraging unlabeled task-specific data through DAPT.

- AdaSent resolves the conflict between DAPT and SEPT and the efficiency issue of the sequential execution of both training procedures, by combining them in a modular fashion without sacrificing the performance.

## 2 Related Work

### 2.1 Text Classification with Sentence Embeddings

Transformer-based (Vaswani et al., 2017) Pre-trained Language Models (PLMs) (Devlin et al., 2019; Liu et al., 2019; Sanh et al., 2019) can be fine-tuned to build sentence embedding models (Reimers and Gurevych, 2019). Since the original goal of training sentence embeddings is to better model the sentence similarity for applications such as dense retrieval and sentence cluster-

ing (Reimers and Gurevych, 2019), their usage is less explored in text classification. Though frozen sentence embeddings can directly serve as input features in text classification (Perone et al., 2018; Piao, 2021), the performance is limited compared to standard full fine-tuning of PLMs (Kumar et al., 2022). To compensate this performance loss, Patel et al. (2021) concatenate encodings from various Sentence Transformers to form semantically richer sentence representations, achieving results comparable to standard fine-tuning, but at the cost of slower inference. More recently, SetFit (Tunstall et al., 2022) significantly improves the few-shot classification by contrastively fine-tuning a pre-trained sentence-embedding model before training a classification head. Despite efficiently utilizing the limited labeled samples, SetFit does not leverage the abundant in-domain unlabeled data that can provide more domain knowledge for the task.

### 2.2 Few-Shot Text Classification

Large language models can perform few-shot classification through ICL with task-specific prompts consisting of a few labeled examples (Brown et al., 2020). Though it avoids any gradient update, ICL relies on large model sizes for good performance, which makes inference costly. Prompt-based fine-tuning, on the other hand, can work with smaller models (Schick and Schütze, 2021; Tam et al., 2021; Gao et al., 2021a). Parameter Efficient Fine-Tuning (PEFT) can further reduce the training cost by fine-tuning a much smaller module in a frozen PLM (Houlsby et al., 2019; Li and Liang, 2021; Hu et al., 2022; Karimi Mahabadi et al., 2022; He et al., 2022; Liu et al., 2022). As an alternative way to employ task instructions, Su et al. (2023) train domain- and task-aware text embeddings by prepending instructions to the input text. In contrast to these methods, SetFit and our approach not only require a smaller model size, but also eliminate the need for prompts or instructions, which can introduce large variance and should be carefully designed (Perez et al., 2021).

### 2.3 Domain Adaptation of Language Models

One typical way for creating domain-specific language models is pre-training through Masked Language Modelling on in-domain corpora, either continuously (Gururangan et al., 2020) or from-scratch (Lee et al., 2019). An alternative is adapting the tokenizer to accommodate domain-specific vocabulary (Sachidananda et al., 2021; Yao et al., 2021).

For sentence embedding models specifically, domain adaptation is usually done through unsupervised training with novel objectives (Wang et al., 2021; Liu and Yang, 2022) or in-domain data generation (Wang et al., 2022), mainly for the similarity or relevance estimation tasks. However, supervised sentence embedding training with general-domain data (SEPT) is always required *after* the unsupervised domain-specific training phase (DAPT) to achieve optimal performance (Wang et al., 2021). Our proposed method is inspired by the idea of disentangling domain adaptation and the downstream relevance estimation task via PEFT in Zhan et al. (2022). In the present study, we show that PEFT can also be used to decouple DAPT and SEPT for few-shot classification tasks.

## 2.4 Semi-Supervised Text Classification

Unsupervised data can be incorporated in various ways to improve few-shot classification. While the DAPT approaches in subsection 2.3 allow the model to learn domain-specific features in a task-agnostic way, other semi-supervised methods typically propagate task information from labeled data to unlabeled data through pseudo labeling. The pseudo-labeled data are either used for self-training (Schick and Schütze, 2021) or consistency training (Xie et al., 2020). All these approaches can also be combined to enable more efficient use of unlabeled data (Li et al., 2021b; Chen et al., 2021; Zhao and Yao, 2022). In our experiments, we found that simple self-training using the same data for DAPT can further improve the performance of AdaSent.

## 3 Background

### 3.1 SetFit

SetFit (Tunstall et al., 2022) is a two-step training procedure based on pre-trained sentence-embedding Transformer models for few-shot sentence classification. In the sentence-embedding fine-tuning step, positive and negative sentence pairs are generated from few-shot labeled sentences as follows: Pairs consisting of sentences from the same class are labeled positively with a score of 1 and pairs of sentences from different classes are assigned a negative score of 0. These generated pairs are used to fine-tune the sentence-embedding model with the Cosine Similarity Loss:

$$L_{\text{cosine}} = \|y - \text{cos\_sim}(u, v)\|_2,$$

where $u, v \in \mathbb{R}^D$ are the $D$-dimensional sentence embeddings of two sentences respectively and $y \in \{0, 1\}$ is the pair label. This aims to push instances of the same classes closer together in the representation space and those from different classes further apart, thereby clustering sentences according to their class labels to provide a clearer decision boundary for the classifier training later. In the second step, the Transformer is frozen to embed the original few-shot sentences. These sentence embeddings are used as input features to train a simple Logistic Regression (Cox, 1958) classification head.

### 3.2 Sentence Embedding Pre-Training (SEPT)

As will be shown in subsection 6.2, the success of SetFit heavily relies on SEPT. This is because the averaged word representations or the [CLS] representation from a PLM cannot capture the sentence semantics well without further training with sentence-level objectives (Reimers and Gurevych, 2019). The purpose of sentence-embedding pre-training is to train universal semantic representations that can be fine-tuned for different downstream tasks, e.g. in SetFit. Unlike SetFit, sentences with similar meaning are brought closer together in SEPT, while those with dissimilar meanings are pushed apart. Sentence pairs for this kind of contrastive training are typically obtained from Natural Language Inference (NLI, Bowman et al., 2015; Williams et al., 2018) or paraphrase datasets in the general domain. Sentence pairs labeled as "entailment" or "paraphrase" in the original datasets are used as positive pairs, i.e. sentences with similar meaning, in SEPT. The Multiple-Negative Ranking Loss (MNRL, Henderson et al., 2017) with in-batch negatives is usually applied for training:

$$L_{\text{MNRL}} = -\frac{1}{K} \sum_{i=1}^{K} \log \frac{e^{\text{cos\_sim}(x_i, y_i)}}{\sum_{j=1}^{K} e^{\text{cos\_sim}(x_i, y_j)}},$$

where $\{(x_i, y_i)\}_{i=1}^{K}$ are a batch of $K$ positive sentence pairs.

### 3.3 Domain-Adapted Sentence Embeddings

The definition of sentence similarity varies from domain to domain, but labeled data for SEPT are usually expensive to obtain in specialized domains. Wang et al. (2021) found that domain-adapted sentence embedding models can be trained following a two-stage recipe: first doing unsupervised DAPT

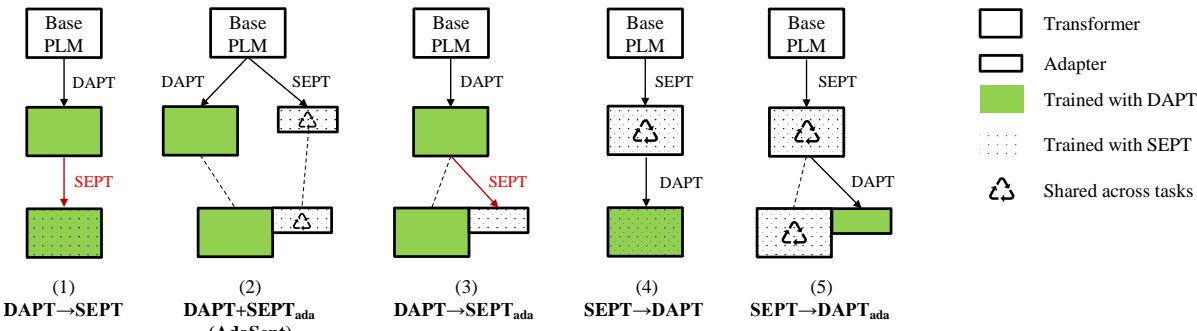

Figure 2: Five ways to combine Domain-Adaptive Pre-Training (DAPT) and Sentence Embedding Pre-Training (SEPT). An arrow pointing from a Transformer to an adapter means the adapter is trained on that Transformer. A dashed line means simple module assembly without any parameter tuning. ♺ marks trained parameters that are reusable and shared across downstream tasks. In contrast, all SEPT training starting from a DAPT Transformer (red arrows) must be repeated on every downstream task.

(e.g. MLM) on the domain-specific corpus, then applying supervised SEPT in the general domain (Figure 2 (1)). With this training order, if we want to train models for various domains, the same second stage has to be repeated for every domain, although it does not involve any domain-specific data. Such computational overhead cannot be avoided by simply reversing the order of the two training stages (Figure 2 (4)), since it has been shown in previous work that DAPT after the generic sentence embedding training has a negative impact on the downstream performance (Wang et al., 2021).

## 4 Method

As illustrated in Figure 1, our method for few-shot classification with domain-adapted sentence embeddings consists of three parts of training: (1) DAPT on the base PLM with task-specific unlabeled data, (2) SEPT on an adapter module with labeled sentence pairs from the general domain and (3) SetFit on the whole architecture (i.e. both the PLM and the adapter) with few-shot labeled data.

In the first part, specifically, we continue to train a base PLM like DistilRoBERTa on unlabeled target task data with the MLM loss to learn domain-specific language knowledge. In another separate procedure, SEPT is done by tuning an adapter on a frozen base Transformer (the same PLM as in DAPT) without any domain adaptation. Once the domain-independent sentence encoding adapter is trained, it can be easily inserted into different DAPT models, ready for the few-shot classification task learning via SetFit in the third part.

Compared to the previous approach described in subsection 3.3, AdaSent is more efficient for three

| Data | All | Paraphrase | NLI+SC+SE | NLI |
|---|---|---|---|---|
| Data Size | 1B | 86M | 0.6M | 0.3M |
| Accuracy | 68.6 | 70.0 | 70.0 | 68.8 |

Table 1: SetFit accuracy on the MTEB classification tasks (see subsection 5.3) of sentence embedding models trained on different SEPT datasets without domain adaption. All and Paraphrase stand for the *all-distilroberta-v1*[3] and the *paraphrase-distilroberta-base-v2*[4], respectively.

reasons. Most significantly, our SEPT adapter is trained only once and shared across various downstream classification tasks, avoiding the overhead of repeating SEPT on new DAPT-ed models. Moreover, AdaSent allows for the independent execution of DAPT and SEPT, eliminating the need for sequential training. Therefore, they can be run concurrently in parallel to save training time. Lastly, training an adapter instead of the full model in SEPT reduces the number of trainable parameters.

Given the extensive number of experiments in this study, we use a mixture of three datasets for SEPT, dubbed **NLI+SC+SE**, consisting SNLI (Bowman et al., 2015) + MultiNLI (Williams et al., 2018), Sentence Compression (Filippova and Altun, 2013) and StackExchange duplicate questions, for the sake of simplicity. This is a much smaller subset of the 1 billion sentence pairs on which the popular off-the-shelf sentence embedding models[5] are pre-trained. We found that these three SEPT datasets transfer the best for the downstream clas-

[3]https://huggingface.co/sentence-transformers/all-distilroberta-v1

[4]https://huggingface.co/sentence-transformers/paraphrase-distilroberta-base-v2

[5]https://huggingface.co/sentence-transformers

sification tasks[6], and are adequate to train a model that performs on par with or even better than off-the-shelf models as shown in Table 1.

## 5 Experimental Setup

### 5.1 Models

We experiment with three baselines and five types of domain-adapted sentence embedding models. All of these models serve as the sentence encoder in the SetFit for the few-shot classification tasks. The baselines are: (1) **Base**, the base PLM without any DAPT or SEPT; (2) **SEPT**, with only SEPT on the base PLM, which is also the default encoder in the original SetFit work; (3) **DAPT**, a domain-adapted PLM, i.e. the Base model continuously pre-trained on the in-domain corpus without SEPT. We also experiment with five variations of domain-adapted sentence embedding models, which differ in the way SEPT and DAPT are combined (Figure 2). In detail, they are: (1) **DAPT→SEPT**, created through DAPT followed by SEPT on the full Transformer parameters without adapter; (2) **DAPT+SEPT$_{ada}$** is our AdaSent model; (3) **DAPT→SEPT$_{ada}$** differs from AdaSent in the training of the SEPT adapter, which is trained on the DAPT model instead of the base PLM; (4) **SEPT→DAPT** reverses the training order of (1), namely doing DAPT after SEPT; (5) **SEPT→DAPT$_{ada}$** trains a DAPT adapter on a frozen SEPT model. It requires the shortest training time, since it avoids any update of the Transformer parameters.

### 5.2 Training Details

We use DistilRoBERTa as the base PLM in our main experiments. Additional results on Distil-BERT are reported in the Appendix D. We set the maximum sequence length to 512. We do not tune the hyperparameters and keep them the same for all downstream tasks. If not stated otherwise, the default setting in the used libraries (cf. Appendix A) is applied. For DAPT with MLM in the main experiments, we train for a fixed number of 2344 steps[7] with a batch size of 256. When using PEFT methods for DAPT, we keep the same batch size and number of steps, but with a larger learning rate of $1e-4$. For SEPT, we train with a batch size of 64 for 1 epoch; the learning rates are 2e-5 and 1e-4 for

full and parameter-efficient training, respectively. For parameter-efficient training, a parallel adapter (He et al., 2022) is used by default. We also provide results of three other different PEFT methods: bottleneck adapter (Houlsby et al., 2019; Pfeiffer et al., 2020), LoRA (Hu et al., 2022) and prefix-tuning (Li and Liang, 2021).

In a separate experiment (subsection 6.1), we compare, on models DAPT, DAPT→SEPT and SEPT→DAPT, three objectives for DAPT: MLM, TSDAE (Wang et al., 2021) and SimCSE (Gao et al., 2021b). The latter two are designed for unsupervised sentence embedding learning, representing two mainstream training objectives for this task: denoising autoencoding and contrastive learning, respectively. For all three objectives, we train on the unlabeled dataset for 3 epochs. The batch sizes are 8, 8, 64 and the learning rates are 5e-5, 3e-5 and 1e-2, respectively. We only use NLI data in SEPT here for simplicity. The same setting is applied for the experiment in subsection 6.5.

For each downstream classification task, we do SetFit on all the models with 8-shot labeled data per class for 1 epoch. The default classification head in SetFit is Logistic Regression.

### 5.3 Evaluation

We evaluate the models on 17 classification tasks, an overview of which is provided in Table 2. These include 11 datasets from the MTEB (Massive Text Embedding Benchmark, Muennighoff et al., 2023). For datasets that contain multilingual data, we only use the English subset in this work. Since most of the MTEB tasks are from the general domain, we add another six tasks for domain-specific cases, including Adverse Drug Events Binary Classification (Gurulingappa et al., 2012) and PubMed RCT (Dernoncourt and Lee, 2017) from the biomedical domain, LEDGAR (Tuggener et al., 2020; Chalkidis et al., 2022) from the legal domain, as well as Financial PhraseBank (Malo et al., 2014), Twitter Financial News Sentiment[8] and Twitter Financial News Topic[9] from the financial domain.

For each task, we sample 8 shots per class from the training set as the labeled data for SetFit and treat the whole original training set as the unlabeled data for DAPT. We run SetFit five times with different random seeds, which correspond to five

---

[6]See Appendix B for results of individual SEPT datasets.
[7]This corresponds to 3 epochs on the largest training set in our evaluation datasets.

[8]https://huggingface.co/datasets/zeroshot/twitter-financial-news-sentiment
[9]https://huggingface.co/datasets/zeroshot/twitter-financial-news-topic

| Dataset | Abbr. | # Train | # Class | Seq. len. (words) | Description |
|---|---|---|---|---|---|
| *MTEB classification* | | | | | |
| Amazon Counterfactual | AC | 4018 | 2 | 20 | Amazon customer reviews labeled as counterfactual or not. |
| Banking77 | BANK | 10003 | 77 | 11 | Banking querys with corresponding intents. |
| Amazon Massive Intent | AMI | 11514 | 60 | 6 | Amazon Alexa utterances with associated intent. |
| Amazon Massive Scenario | AMS | 11514 | 18 | 6 | Amazon Alexa utterances with theme. |
| MTOP Intent | MI | 15667 | 113 | 7 | Task-oriented dialog utterances with intent. |
| MTOP Domain | MD | 15667 | 11 | 7 | Task-oriented dialog utterances with domain. |
| Emotion | EMO | 16000 | 6 | 19 | Twitter messages with basic emotion type. |
| IMDb | IMDB | 25000 | 2 | 233 | Movie reviews as positive or negative. |
| Twitter Sentiment Extraction | TSE | 27481 | 3 | 12 | Tweet sentiment classification as neutral, positive or negative. |
| Toxic Conversation | TC | 50000 | 2 | 51 | Comments from the Civil Comments platform as toxic or not. |
| Amazon Reviews Multi | ARM | 200000 | 5 | 38 | Amazon reviews with 1-5 stars. |
| *Domain-specific tasks* | | | | | |
| Financial PhraseBank | FPB | 3876 | 3 | 23 | Financial news headlines with the view of a retail investor. |
| Twitter Financial News Sentiment | TFNS | 8588 | 3 | 12 | Finance-related tweets with their sentiment. |
| Twitter Financial News Topic | TFNT | 15291 | 20 | 18 | Finance-related tweets with their topic. |
| Adverse Drug Events | ADE | 18812 | 2 | 19 | Classify if a sentence is ADE-related or not. |
| PubMed RCT | RCT | 176642 | 5 | 27 | PubMed abstract sentences with their role in the abstract. |
| LEDGAR | LED | 60000 | 100 | 114 | Contract provisions with their main topic. |

Table 2: Overview of the evaluation datasets. All tasks are multi-class classification. From the training set, only 8 labeled shots per class are used for SetFit. The whole training set is used in DAPT without labels. Examples from each dataset can be found in Appendix F.

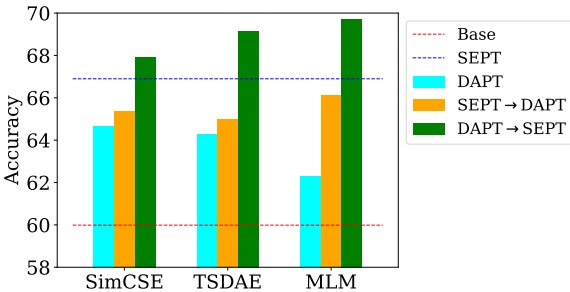

Figure 3: Averaged accuracy on 17 datasets of different DAPT training objectives (SimCSE, TSDAE, MLM) and different training strategies (without, before or after SEPT). Results on individual datasets are in Table 11.

different sets of few-shot samples. We report the average accuracy on the test set of each dataset over the five runs.

## 6 Results

### 6.1 Training Order and DAPT Objectives

In our first experiment, we compare two training orders: SEPT→DAPT and DAPT→SEPT, and three DAPT objectives: MLM, TSDAE and SimCSE. The results are shown in Figure 3.

Regarding the training order, DAPT→SEPT outperforms SEPT→DAPT for all three DAPT objectives. DAPT can enhance the SEPT baseline only when it is performed prior to SEPT, but this setting has the efficiency issue described in subsection 3.3. On the other hand, DAPT has a negative impact on an already pre-trained sentence encoder, because

it may distort the sentence representation space shaped by SEPT. These findings on our classification tasks are consistent with those on the retrieval tasks in Wang et al. (2021).

With the DAPT→SEPT order, MLM achieves the best result among three DAPT objectives, improving the SEPT baseline by around 3 points on average. Although TSDAE has been shown to have a clear advantage in tasks like re-ranking and paraphrase identification (Wang et al., 2021), it turns out to be suboptimal for sentence classification. On the contrary, MLM performs worse than TSDAE and SimCSE when there is no SEPT. We suppose that sentence classification with SetFit requires a good representation of both token- and sentence-level semantics, which are learned through MLM and SEPT respectively in the $DAPT_{MLM}$→SEPT setting. In other settings, either supervised sentence embedding training is absent (only DAPT), or token representation learning is missing (both TSDAE and SimCSE are for sentence representation learning).

### 6.2 Combination of DAPT and SEPT

In this subsection, we present the results of our main experiments on various combination strategies for DAPT and SEPT. The results on the MTEB datasets are reported in Table 3, and those for the domain-specific datasets are in Table 4. AdaSent achieves the best result on 10 out of 17 tasks, outperforming the not domain-adapted SEPT model

| Row No. | Model | AC | BANK | AMI | AMS | MI | MD | EMO | IMDB | TSE | TC | ARM | Avg. |
|---|---|---|---|---|---|---|---|---|---|---|---|---|---|
| | | | | | | *No SEPT* | | | | | | | |
| R1 | Base | 65.9 | 75.7 | 62.0 | 71.0 | 72.8 | 89.4 | 40.2 | 67.7 | 50.9 | 55.2 | 37.3 | 62.6 |
| R2 | DAPT | 69.4 | 80.4 | 70.0 | 79.4 | 80.6 | 94.7 | 37.3 | 74.9 | 55.1 | 48.2 | 41.9 | 66.5 |
| | | | | | | *Full SEPT* | | | | | | | |
| R3 | SEPT (prev. SOTA) | 76.1 | 77.0 | 66.8 | 73.3 | 78.4 | 90.6 | 52.2 | 84.8 | 63.2 | 63.6 | 44.2 | 70.0 |
| R4 | DAPT→SEPT | 73.8 | **80.9** | 73.5 | 79.2 | **83.7** | 94.5 | 54.0 | 86.2 | 63.7 | 63.6 | 46.9 | 72.7 |
| | | | | | | *SEPT on Adapter* | | | | | | | |
| R5 | SEPT$_{ada}$ | 76.0 | 76.9 | 66.0 | 73.6 | 79.4 | 91.4 | 55.3 | 84.6 | 63.1 | 65.4 | 43.8 | 70.5 |
| R6 | DAPT→SEPT$_{ada}$ | **77.9** | 80.7 | **73.8** | 79.3 | 82.9 | 94.7 | **54.7** | 85.6 | 65.0 | **65.5** | 47.0 | **73.4** |
| R7 | AdaSent | **77.9** | 80.6$^\dagger$ | 73.7$^\dagger$ | **80.5**$^\dagger$ | 82.7$^\dagger$ | **95.4**$^\dagger$ | 54.1 | **86.7** | 65.2 | 63.0 | **48.1**$^\dagger$ | **73.4** |
| | | | | | | *DAPT on Adapter* | | | | | | | |
| R8 | SEPT→DAPT$_{ada}$ | 72.8 | 79.5 | 69.8 | 78.0 | 81.0 | 93.7 | 48.3 | 84.4 | 59.2 | 59.4 | 44.9 | 70.1 |

Table 3: Classification accuracy on the MTEB classification tasks. Full SEPT means tuning all the PLM parameters in the sentence embedding pre-training. Best results on each dataset are in **bold**. † marks the cases where AdaSent outperforms SEPT (R5) with a statistical significance level of 0.05.

| Row No. | Model | FPB | TFNS | TFNT | ADE | RCT | LED | Avg. |
|---|---|---|---|---|---|---|---|---|
| | | | | *No SEPT* | | | | |
| R1 | Base | 49.2 | 51.1 | 57.7 | 60.7 | 49.6 | 64.2 | 55.4 |
| R2 | DAPT | 50.3 | 56.3 | 64.8 | 67.8 | 57.8 | 66.7 | 60.6 |
| | | | | *Full SEPT* | | | | |
| R3 | SEPT | 63.0 | 65.0 | 62.2 | 62.3 | 61.5 | 65.6 | 63.3 |
| R4 | DAPT→SEPT | 65.6 | 69.4 | 68.4 | 67.4 | 66.5 | **68.1** | 67.6 |
| | | | | *SEPT on Adapter* | | | | |
| R5 | SEPT$_{ada}$ | 64.2 | 66.1 | 61.4 | 62.8 | 58.7 | 65.9 | 63.2 |
| R6 | DAPT→SEPT$_{ada}$ | 66.1 | **69.9** | 68.5 | 65.8 | 67.4 | 68.0 | 67.6 |
| R7 | AdaSent | **66.4** | 69.8 | 68.6$^\dagger$ | **67.8** | **67.5** | 67.8$^\dagger$ | **68.0** |
| | | | | *DAPT on Adapter* | | | | |
| R8 | SEPT→DAPT$_{ada}$ | 62.7 | 62.0 | 65.2 | 66.7 | 64.0 | 66.8 | 64.6 |

Table 4: Classification accuracy on the domain-specific datasets. Best results on each dataset are in **bold**. † marks the cases where AdaSent outperforms SEPT (R5) with a statistical significance level of 0.05.

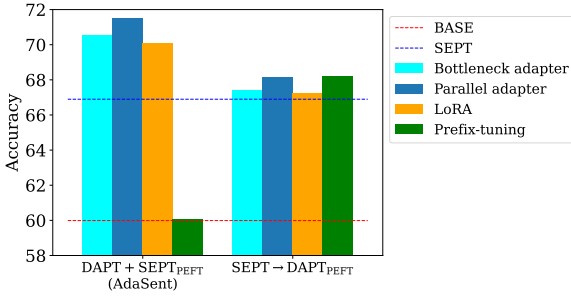

Figure 4: Averaged accuracy of different PEFT methods. SEPT$_{PEFT}$ stands for SEPT on a PEFT module. More detailed results are available in Table 13.

by 3.9 on average on the MTEB datasets, and more prominently, by 4.7 on the datasets in Table 4 with a larger domain shift from the pre-training data. The improvement is statistically significant on 8 datasets, with a significance level of 0.05. Our following analysis will focus on Table 3, while similar trends can be observed in Table 4.

SEPT is crucial to the final accuracy of classification methods based on sentence embeddings like SetFit, though this is not explicitly mentioned in the original SetFit paper (Tunstall et al., 2022). SEPT improves both the Base model (R3 vs. R1) and the DAPT model (R4 vs. R2) by 7.3 and 6.2 points on average, respectively.

By adding a DAPT stage before SEPT, the classification accuracy can be significantly increased by up to 6.7 points (on AMI) and 2.7 points on average (R4 vs. R3). However, as we discussed in subsection 3.3, executing the same SEPT procedure on every DAPT model results in computational inefficiency. As a more efficient alternative, our AdaSent avoids repeating SEPT by sharing a SEPT adapter

across different downstream tasks, while obtaining comparable results without statistically significant difference (R7 vs. R4), except for the AMS dataset, where Adasent is even significantly better than DAPT→SEPT. The comparable performance of DAPT→SEPT$_{ada}$ and AdaSent (R6 vs. R7) proves the viability of decoupling DAPT and SEPT: The SEPT adapter does not have to be trained on a specific DAPT model. Instead of doing SEPT on adapter, we also tried with DAPT on adapter (SEPT→DAPT$_{ada}$), which should be the most efficient method as explained in subsection 5.1. Disappointingly, it can barely improve over SEPT (R8 vs. R3) and is much worse than AdaSent (R8 vs. R7). The reason could be that this setting suffers from the same problem as SEPT→DAPT, as the DAPT phase, despite on an adapter, is still conducted after SEPT.

## 6.3 Comparison of PEFT Methods

We experimented with four different PEFT methods for both SEPT and DAPT (Figure 4). When applied to SEPT in AdaSent, parallel adapter works best on the majority of the datasets (Ta-

| Tunable Parameters | None (0%) | Adapter (4%) | Transformer (96%) | All (100%) |
|---|---|---|---|---|
| Accuracy | 65.0 | 69.0 | 71.2 | 71.5 |

Table 5: Results of tuning subsets of model parameters (marked with relative sizes) in the final SetFit stage of AdaSent. None means only training the logistic regression head.

| | MLM | TSDAE |
|---|---|---|
| DAPT+SEPT$_{ada}$ (AdaSent) | 69.7 | 67.3 |
| DAPT→SEPT | 69.7 | 69.1 |

Table 6: Averaged accuracy of different DAPT objectives in AdaSent and DAPT→SEPT.

ble 13) and on average. Prefix-tuning is significantly worse than the other three methods. This might be due to the fact that the data in our SEPT dataset NLI+SC+SE come from three different tasks, whose properties cannot be compressed into a single prefix. When applied to DAPT in the SEPT→DAPT$_{PEFT}$ setting, their performance exhibits variability across different datasets (Table 13), but none of the four PEFT methods in this setting can beat the AdaSent variants due to the critical drawback of the setting as discussed at the end of subsection 6.2.

### 6.4 Tunable Parameters in SetFit

We tune various subsets of parameters in the SetFit stage of AdaSent and compare the results in Table 5. We found that only updating the adapter parameters is not sufficient. However, tuning only the Transformer backbone leads to almost the same results as tuning all parameters (i.e. Transformer + adapter). This indicates that with only few-shot labeled data, SetFit must at least update the Transformer parameters to achieve good performance, and cannot work well on an adapter as in the case of SEPT, where much more supervised data are available.

### 6.5 Explaining the Success of AdaSent

The success of AdaSent relies on the fact that a SEPT adapter trained on a base PLM can be unproblematically inserted into any domain-adapted version of the same PLM. This might be because in both original pre-training and domain-adaptive pre-training, the PLM parameters are consistently tuned with the MLM objective. This implies that the adapter can generalize to work together with PLM parameters trained on different types of data, from general-language data (e.g. BookCorpus, Zhu

| Self-training | No | Yes |
|---|---|---|
| SEPT | 67.6 | 68.6 (+1.0) |
| DAPT+SEPT$_{ada}$ (AdaSent) | 71.5 | 72.4 (+0.9) |

Table 7: Averaged accuracy of AdaSent and SEPT, w/ or w/o self-training.

| Method | DAPT Steps | Cost (hour) | | | Acc. |
|---|---|---|---|---|---|
| | | SEPT | DAPT | Total | |
| DAPT→SEPT | 0 | | 0.00 | 4.05 | 67.6 |
| | 100 | | 0.44 | 4.49 | 69.8 |
| | 500 | 15 × 0.27 | 2.21 | 6.26 | 70.7 |
| | 1000 | | 4.42 | 8.47 | 71.1 |
| | 2000 | | 8.83 | 12.88 | 71.5 |
| AdaSent | 0 | | 0.00 | 0.17 | 67.9 |
| | 100 | | 0.44 | 0.61 | 69.5 |
| | 500 | 1 × 0.17 | 2.21 | 2.38 | 70.7 |
| | 1000 | | 4.42 | 4.59 | 71.2 |
| | 2000 | | 8.83 | 9.00 | 71.4 |

Table 8: Total training cost on 15 tasks with Distil-RoBERTa as base PLM on a Tesla V100 GPU.

et al., 2015) to domain-specific data, as long as the same MLM objective is used. To verify this idea, we replace the MLM objective with TSDAE in both AdaSent and DAPT→SEPT. As shown in Table 6, using TSDAE instead of MLM in the DAPT stage of AdaSent leads to a substantial decrease of 2.4 points in the classification accuracy, while the performance drop in DAPT→SEPT is relatively marginal (0.6 on average). This supports our hypothesis that the adapter can only generalize to collaborate with PLM parameters that are domain-adapted with the same objective as in the pre-training.

### 6.6 Combining DAPT and Self-Training

Besides DAPT, another major way to utilize the unlabeled data is self-training, which has been shown to be complementary to DAPT (Li et al., 2021b). To integrate self-training into SetFit, we first encode the unlabeled data with the sentence encoder (in our case a DAPT Transformer + SEPT adapter) trained with few-shot labeled data in the contrastive fine-tuning phase. When training the classification head, we iteratively pseudo-label the encoded unlabeled sentences and train with both the pseudo-labeled and the gold-labeled data[10]. In Table 7, we show that self-training can further improve both SEPT and AdaSent's accuracy by 1.0 and 0.9 on average, respectively. These two close improvements reveal that the benefit of self-training is orthogonal to that of AdaSent/DAPT. We leave more complex

---

[10]The training details are available in Appendix G.

approaches of combining AdaSent and self-training for future work.

## 7 Training Cost

Table 8 gives an overview of the training cost for DAPT→SEPT and AdaSent in our experiments. We use a Tesla V100 GPU for training. We leave out IMDB and LED as they have too long sequences (cf. Table 2), thus cannot represent the majority of our tasks.

With AdaSent, SEPT is trained once for 0.17h and the SEPT adapter can be shared across tasks. In contrast, DAPT→SEPT costs 0.27 hours additionally for every task due to its repeated SEPT. In our experiments, we use relatively small-sized data for SEPT. However, the SEPT cost can increase dramatically if much larger training data are used. For example, SEPT on the combination of all datasets in Table 10 for 1 epoch can take 4 hours, resulting in $15 \times 4$ hours for DAPT→SEPT for 15 tasks. For DAPT, we can see that 1000 steps are already sufficient for a substantial improvement in accuracy. In this case, AdaSent takes 4.59 hours for the training on 15 tasks in total, while DAPT→SEPT takes 8.47 hours ($\times 1.85$).

## 8 Conclusion

We introduce an efficient method to obtain domain-adapted sentence embeddings for few-shot classification. We found that SetFit, the previous state-of-the-art approach, can be significantly improved by introducing a simple Domain-Adaptive Pre-Training (DAPT) stage before its Sentence-Embedding Pre-Training (SEPT). However, this DAPT→SEPT approach requires the same SEPT procedure to be done on each DAPT-ed PLM for every domain, resulting in computational inefficiency. We propose a novel approach, AdaSent, to address this issue by storing the SEPT knowledge in an adapter that is trained on an unadapted PLM and insertable into any DAPT-ed PLM. AdaSent matches or surpasses the performance of DAPT→SEPT, while significantly reducing the training cost of SEPT. We attribute the success of AdaSent to the generalization ability of the SEPT adapter to work with PLM parameters trained on data from different domains with a consistent MLM objective.

## Limitations

Since our method is based on SetFit, it inherits some of its limitations. It is, for example, not applicable for sentence pair classification like NLI. In addition, the advantage of SetFit is not significant in classification tasks with too many classes. Moreover, as our method is based on sentence embeddings, its application is limited to sentence classification, unlike other few-shot classification methods that can also handle token-level classification tasks like NER and POS tagging.

Another limitation is associated with the fact that the SEPT adapter in our method can only be inserted into domain-adapted language models with the same unmodified tokenizer and vocabulary as the original base PLM. For DAPT-ed models with a domain-specific tokenizer or vocabulary, we suppose the adapter trained on the original PLM will not be compatible anymore.

## Ethics Statement

Our experiments use publicly available datasets and benchmarks for training and evaluation, which are commonly used in the field of NLP. No personal information or sensitive data are involved in our work. Existing biases in the datasets or pre-trained models can still be relevant concerns, since we do not specifically focus on mitigating them in the current work.

## Acknowledgements

This work has been funded by HUAWEI Technologies (Ireland) Co., Ltd. and by the German Federal Ministry of Education and Research (BMBF) under the promotional reference 13N15897 (MISRIK).

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

## A Implementation

See Table 9 for the implementation of methods used in this work.

## B Experiment with SEPT Datasets

We experiment with different SEPT datasets[11] to check their transferability to downstream tasks Table 10. On average, AllNLI, SentenceCompression and StackexchangeDuplicateQuestions are the top three datasets. The similarity between the SEPT data and the downstream data seems to have an influence on the performance. For example, QA-related data (YahooAnswersTitleAnswer, StackexchangeDuplicateQuestions and YahooAnswersQuestionAnswer) are especially beneficial for the classification tasks involving user utterances in dialogues (BANK, AMI, AMS, MI, MD). Given this observation, one might want to search for the optimal SEPT datasets depending on certain types of classification tasks. Our adapter-based method enables efficient SEPT, which helps to ease the data selection.

## C DAPT Objectives and Training Order

Results on individual datasets are listed in Table 11.

## D Results on DistilBERT

We report the results on DistilBERT in Table 12. Similar to DistilRoBERTa, DAPT with MLM

| Method | Used Implementation |
|---|---|
| PEFT | https://github.com/adapter-hub/adapter-transformers |
| TSDAE | https://github.com/UKPLab/sentence-transformers |
| SEPT | https://github.com/UKPLab/sentence-transformers |
| SimCSE | https://github.com/princeton-nlp/SimCSE |
| MLM | https://github.com/huggingface/transformers/blob/main/examples/pytorch/language-modeling/run_mlm_no_trainer.py |
| SetFit | https://github.com/huggingface/setfit |

Table 9: Implementation used in this work.

(DAPT→SEPT and DAPT+SEPT$_{ada}$) improves the performance of SEPT by around 3 points on average. Replacing full SEPT with SEPT adapter causes a slight drop of around 0.5 in the classification accuracy. Interestingly, without any supervised sentence embedding pre-training, DAPT itself can outperform SEPT on some datasets (AC, ADE, LED).

## E PEFT results

Results on individual datasets when using different PEFT methods as discussed in subsection 6.3 in our AdaSent method (DAPT+SEPT$_{PEFT}$) and SEPT→DAPT$_{PEFT}$ are shown in Table 13.

## F Evaluation Datasets

Table 14 provides examples from each evaluation dataset.

## G Self-Training Setting

In the SetFit phase, we contrastively fine-tune the sentence embedding model with the few-shot data as before (subsection 3.1), but replace the normal Logistic Regression fitting with self-training on both labeled and unlabeled data. For this, we use the SelfTrainingClassifier from scikit-learn[12] with 10 iterations and a threshold of 0.9. At each iteration, the classifier predicts the label of the unlabeled data. The pseudo-labeled data with a confidence score higher than the threshold are used to augment the training data in the next iteration.

---

[11]See https://www.sbert.net/examples/training/paraphrases/README.html for information of the datasets.

[12]https://scikit-learn.org/stable/modules/generated/sklearn.semi_supervised.SelfTrainingClassifier.html

| SEPT Data | AC | BANK | AMI | AMS | MI | MD | EMO | IMDB | TSE | TC | ARM | Avg. |
|---|---|---|---|---|---|---|---|---|---|---|---|---|
| AllNLI | 65.5 | **75.5** | 63.5 | 72.1 | 74.2 | 90.2 | 48.8 | 84.9 | 62.7 | 64.8 | 43.5 | 67.8 |
| SentenceCompression | **74.0** | 74.9 | 61.5 | 72.9 | 74.9 | 90.6 | **52.4** | 83.8 | 60.3 | 58.7 | 42.2 | **67.9** |
| SimpleWiki | 65.4 | 74.3 | 59.4 | 71.1 | 70.8 | 89.3 | 45.3 | 83.5 | 59.7 | 63.8 | 41.9 | 65.9 |
| Altlex | 68.4 | 74.5 | 59.6 | 71.1 | 72.0 | 89.2 | 45.6 | 81.8 | 57.7 | 62.6 | 41.6 | 65.8 |
| QuoraDuplicatesTriplets | 73.6 | 75.3 | 60.9 | 71.1 | 75.0 | 89.5 | 44.6 | 81.0 | 58.0 | 61.8 | 42.0 | 66.6 |
| CocoCaptions | 58.4 | 74.3 | 58.7 | 71.2 | 71.6 | 89.6 | 45.5 | 60.3 | 51.2 | 58.4 | 37.7 | 61.5 |
| Flickr30kCaptions | 58.0 | 74.1 | 59.5 | 71.2 | 73.8 | 89.4 | 45.1 | 60.0 | 52.3 | **66.4** | 36.8 | 62.4 |
| YahooAnswersTitleQuestion | 69.0 | 75.2 | 60.5 | 72.4 | 75.5 | 90.6 | 46.3 | 83.4 | 55.1 | 58.6 | 41.6 | 66.2 |
| YahooAnswersTitleAnswer | 71.5 | 75.1 | 61.2 | **73.7** | 75.5 | 90.9 | 44.9 | 80.9 | 55.6 | 52.1 | 41.5 | 65.7 |
| StackexchangeDuplicateQuestions | 72.0 | 75.1 | **64.0** | 73.6 | **77.9** | 90.2 | 46.1 | 77.2 | 58.2 | 60.6 | 43.1 | 67.1 |
| YahooAnswersQuestionAnswer | 67.3 | 75.0 | 61.2 | 73.4 | 75.4 | 90.8 | 45.3 | 81.5 | 52.3 | 62.7 | 41.2 | 66.0 |

Table 10: Results on MTEB tasks of SEPT model trained on different datasets. The best scores are marked in bold and second best with underline. We sample 100K instances from each SEPT dataset and train for 1 epoch.

| Model | AC | BANK | AMI | AMS | MI | MD | EMO | IMDB | TSE | TC | ARM | FPB | TFNS | TFNT | ADE | RCT | LED | Avg. |
|---|---|---|---|---|---|---|---|---|---|---|---|---|---|---|---|---|---|---|
| | | | | | | | *TSDAE* | | | | | | | | | | | |
| DAPT | 74.1 | 77.6 | 64.4 | 76.0 | 77.4 | 93.0 | 46.3 | 78.4 | 53.7 | 49.6 | 44.8 | 51.1 | 55.7 | 61.6 | 62.5 | 58.8 | 67.8 | 64.3 |
| SEPT→DAPT | **79.5** | 77.5 | 66.8 | 76.4 | 79.9 | 93.2 | 46.1 | 79.5 | 50.6 | 47.6 | 47.3 | 58.4 | 53.2 | 63.0 | 60.0 | 57.6 | **68.3** | 65.0 |
| DAPT→SEPT | 72.9 | 77.3 | **67.9** | 76.3 | 79.8 | **93.4** | 52.0 | 85.6 | 63.6 | 62.1 | 48.9 | 66.6 | 63.9 | 66.2 | 65.3 | 65.4 | 68.1 | 69.1 |
| | | | | | | | *SimCSE* | | | | | | | | | | | |
| DAPT | 71.2 | 75.3 | 61.9 | 73.0 | 75.6 | 89.9 | 46.8 | 78.5 | 59.2 | 57.2 | 44.5 | 58.8 | 57.3 | 60.5 | 61.3 | 63.6 | 64.3 | 64.6 |
| SEPT→DAPT | 66.2 | 75.4 | 63.3 | 74.0 | 78.8 | 89.8 | 44.4 | 66.9 | 59.1 | **69.9** | 44.7 | 64.9 | 61.7 | 60.2 | 58.3 | **68.2** | 65.5 | 65.4 |
| DAPT→SEPT | 71.4 | 76.0 | 65.4 | 73.9 | 78.8 | 91.9 | 48.4 | 84.7 | **64.1** | 66.4 | 46.9 | 66.9 | 62.5 | 62.3 | 62.7 | 66.1 | 66.0 | 67.9 |
| | | | | | | | *MLM* | | | | | | | | | | | |
| DAPT | 61.8 | 76.6 | 61.8 | 75.7 | 77.0 | 93.2 | 38.0 | 64.8 | 52.7 | 60.3 | 44.2 | 53.2 | 58.3 | 63.9 | 63.9 | 46.4 | 67.3 | 62.3 |
| SEPT→DAPT | 73.2 | 76.4 | 65.0 | 75.6 | 79.0 | 92.8 | 49.1 | 79.3 | 58.3 | 51.3 | 48.8 | 55.9 | 61.5 | 65.9 | 64.7 | 59.2 | 67.9 | 66.1 |
| DAPT→SEPT | 72.7 | **78.0** | 67.4 | 77.0 | 82.4 | 93.7 | 49.9 | 85.5 | 63.9 | 65.3 | 50.9 | 66.6 | 63.8 | 66.8 | 66.5 | 66.9 | 67.7 | 69.7 |
| | | | | | | | *Baselines* | | | | | | | | | | | |
| SEPT | 70.2 | 75.5 | 64.5 | 73.6 | 77.4 | 90.6 | 51.8 | 84.2 | 63.3 | 61.8 | 43.3 | 65.5 | 60.8 | 61.8 | 64.0 | 64.3 | 64.6 | 66.9 |
| Base | 65.9 | 75.2 | 60.6 | 71.0 | 73.9 | 89.4 | 40.3 | 68.0 | 50.9 | 55.6 | 37.6 | 48.5 | 50.8 | 57.8 | 60.9 | 49.2 | 64.2 | 60.0 |

Table 11: Comparison of different DAPT objectives and training order of DAPT and SEFT. The best scores are marked in bold and second best with underline. Note that the training settings of DAPT here is different from that in Table 3 and Table 4: We do DAPT for 3 epochs instead of a fixed-number of steps.

| Model | AC | BANK | AMI | AMS | MI | MD | EMO | IMDB | TSE | TC | ARM | FPB | TFNS | TFNT | ADE | RCT | LED | Avg. |
|---|---|---|---|---|---|---|---|---|---|---|---|---|---|---|---|---|---|---|
| | | | | | | | *No SEPT* | | | | | | | | | | | |
| Base | 74.1 | 73.4 | 61.9 | 70.6 | 75.6 | 88.2 | 35.3 | 64.5 | 49.2 | 58.0 | 37.9 | 49.6 | 43.4 | 52.6 | 64.7 | 53.2 | 64.3 | 59.8 |
| DAPT | **82.1** | 79.6 | 70.2 | 79.2 | 82.9 | 95.1 | 40.1 | 78.3 | 56.1 | 50.6 | 45.2 | 53.8 | 53.2 | 65.6 | **72.4** | 65.3 | 67.3 | 66.9 |
| | | | | | | | *Full SEPT* | | | | | | | | | | | |
| SEPT | 72.6 | 75.7 | 67.5 | 74.3 | 79.7 | 91.9 | 48.4 | 80.7 | 63.8 | 63.9 | 42.8 | 61.1 | 62.1 | 58.2 | 63.9 | 58.6 | 65.6 | 66.5 |
| DAPT→SEPT | 75.6 | 79.7 | **73.9** | 80.4 | 83.9 | 94.9 | 50.9 | 83.4 | 63.8 | **64.4** | 46.1 | 62.2 | 64.6 | **66.3** | 65.9 | 64.0 | 67.3 | 69.8 |
| | | | | | | | *SEPT on Adapter* | | | | | | | | | | | |
| SEPTada | 75.5 | 75.2 | 66.3 | 73.7 | 78.5 | 91.4 | 50.0 | 78.9 | 63.5 | 58.2 | 42.1 | 63.4 | 61.8 | 55.9 | 62.3 | 58.2 | 65.3 | 65.9 |
| AdaSent | 80.6 | **79.8** | 72.1 | 80.3 | 82.4 | **95.7** | **51.6** | 82.2 | 62.8 | 56.6 | **47.1** | 60.4 | **66.2** | 63.4 | 64.9 | 65.8 | 67.3 | 69.4 |

Table 12: Results on DistilBERT. Best scores are in bold.

| PEFT method | AC | BANK | AMI | AMS | MI | MD | EMO | IMDB | TSE | TC | ARM | FPB | TFNS | TFNT | ADE | RCT | LED | Avg. |
|---|---|---|---|---|---|---|---|---|---|---|---|---|---|---|---|---|---|---|
| | | | | | | | DAPT+SEPT_PEFT (AdaSent) | | | | | | | | | | | |
| Bottleneck adapter | 76.3 | 80.7 | 73.0 | 80.2 | 82.2 | 95.3 | 51.5 | **87.4** | 64.2 | 58.6 | 48.0 | 65.0 | 66.7 | 68.6 | 65.2 | **68.7** | 68.2 | 70.6 |
| Parallel adapter | 77.9 | 80.6 | **73.7** | 80.5 | 82.7 | 95.4 | **54.1** | 86.7 | 65.2 | 63.0 | 48.1 | 66.4 | 69.8 | 68.6 | 67.8 | 67.5 | 67.8 | **71.5** |
| LoRA | **79.0** | 80.7 | 72.9 | 79.6 | 82.0 | 94.6 | 52.9 | 85.1 | 63.1 | 60.1 | 46.3 | 63.5 | 64.7 | 66.9 | 67.1 | 65.7 | 67.5 | 70.1 |
| Prefix-tuning | 53.1 | 80.1 | 69.8 | 78.5 | 80.3 | 94.1 | 41.8 | 62.9 | 49.0 | 46.7 | 36.0 | 41.8 | 47.9 | 65.3 | 58.8 | 47.6 | 67.3 | 60.0 |
| | | | | | | | SEPT→DAPT_PEFT | | | | | | | | | | | |
| Bottleneck Adapter | 76.4 | 77.5 | 67.0 | 74.6 | 79.6 | 91.5 | 50.3 | 82.9 | 61.4 | 58.0 | 43.5 | 60.7 | 63.7 | 63.6 | 64.6 | **65.1** | 65.8 | 67.4 |
| Parallel Adapter | 72.8 | 79.5 | 69.8 | 78.0 | 81.0 | 93.7 | 48.3 | 84.4 | 59.2 | 59.4 | **44.9** | 62.7 | 62.0 | 65.2 | 66.7 | 64.0 | 66.8 | 68.1 |
| LoRA | 77.5 | 77.3 | 66.7 | 73.6 | 78.5 | 91.2 | 51.0 | 83.8 | 63.2 | 58.9 | 43.6 | 61.1 | 60.3 | 63.6 | 63.3 | 64.2 | 65.5 | 67.3 |
| Prefix-tuning | **78.9** | 77.3 | 66.0 | 72.6 | 78.6 | 90.5 | 52.9 | 84.7 | 62.6 | 63.7 | 44.2 | 64.5 | 67.7 | 62.7 | 63.8 | 63.0 | 65.8 | 68.2 |

Table 13: Results on individual datasets of different PEFT methods for DAPT+SEPT_PEFT (AdaSent) and SEPT→DAPT_PEFT. Best scores are in bold for both models.

| Dataset | Abbr. | Text | Label |
|---------|-------|------|-------|
| | | *MTEB classification* | |
| Amazon Counterfactual (O'Neill et al., 2021) | AC | In person it looks as though it would have cost a lot more. | counterfactual |
| Banking77 (Casanueva et al., 2020) | BANK | I am still waiting on my card? | card_arrival |
| Amazon Massive Intent (FitzGerald et al., 2023) | AMI | wake me up at nine am on friday | alarm_set |
| Amazon Massive Scenario (FitzGerald et al., 2023) | AMS | wake me up at nine am on friday | alarm |
| MTOP Intent (Li et al., 2021a) | MI | Has Angelika Kratzer video messaged me? | GET_MESSAGE |
| MTOP Domain (Li et al., 2021a) | MD | Has Angelika Kratzer video messaged me? | messaging |
| Emotion (Saravia et al., 2018) | EMO | ive been feeling a little burdened lately wasnt sure why that was | sadness |
| Imdb (Maas et al., 2011) | IMDB | I rented I AM CURIOUS-YELLOW from my video store because of all the controversy that surrounded it when it was first released in 1967. I also heard that at first it was seized by U.S. customs if it ever tried to enter this country, therefore being a fan of films considered "controversial" I really had to see this for myself.

The plot is centered around a young Swedish drama student named Lena who wants to learn everything she can about life. [...] I AM CURIOUS-YELLOW is a good film for anyone wanting to study the meat and potatoes (no pun intended) of Swedish cinema. But really, this film doesn't have much of a plot. | negative |
| Twitter Sentiment Extraction | TSE | I`d have responded, if I were going | neutral |
| Toxic Conversation | TC | theres not enough going on around here for air service none want to waste there time on this town | not toxic |
| Amazon Reviews Multi (McAuley and Leskovec, 2013) | ARM | I received my first order of this product and it was broke so I ordered it again. The second one was broke in more places than the first. I can't blame the shipping process as it's shrink wrapped and boxed. | 0 |
| | | *Domain-specific tasks* | |
| Financial PhraseBank | FPB | With the new production plant the company would increase its capacity to meet the expected increase in demand and would improve the use of raw materials and therefore increase the production profitability . | Positive |
| Twitter Financial News Sentiment | TFNS | Grubhub gains a bear on margin view | Bearish |
| Twitter Financial News Topic | TFNT | Analysts reveal the top stocks with 'significant upside potential' heading into earnings https://t.co/lfaLK3nwAz | Analyst Update |
| Adverse Drug Events | ADE | Intravenous azithromycin-induced ototoxicity. | Related |
| PubMed RCT | RCT | Outcome measures included pain reduction and improvement in function scores and systemic inflammation markers . | Methods |
| LEDGAR | LED | Except as otherwise set forth in this Debenture, the Company, for itself and its legal representatives, successors and assigns, expressly waives presentment, protest, demand, notice of dishonor, notice of nonpayment, notice of maturity, notice of protest, presentment for the purpose of accelerating maturity, and diligence in collection. | Waivers |

Table 14: Examples from evaluation datasets.