# OpenReview forum: "AdaSent: Efficient Domain-Adapted Sentence Embeddings for Few-Shot Classification"
_EMNLP/2023/Conference — EMNLP 2023 Main_

### Official Review · Reviewer_jeTA · 2023-08-01

**Soundness:** 4

**Excitement:**

4: Strong: This paper deepens the understanding of some phenomenon or lowers the barriers to an existing research direction.

**Paper Topic And Main Contributions:**

This paper discusses an adapter technique that avoids the need to finetune on the sentence embedding objective for each domain-adapted language model. The proposed approach is efficient in that the sentence embedding objective only needs to be trained once but can be used with all domain-adapted models. The approach also leads to slight improvements over full finetuning.

**Reasons To Accept:**

Important contribution: The ability to train the sentence embedding adapter only once and use it with the other domain-adapted LMs is significant. Performing domain-specific adaptation is also a good contribution but not a surprising result.

Extensive experiments: The authors performed experiments across many datasets, giving strong evidence for their approach.

Clear writing: The paper is clear and organized well. The authors explain their reasoning for improved performance of their approach over full finetuning (with additional experiments), which I thought was necessary for the paper since it is a somewhat counterintuitive finding.

Citation of previous work: The authors do a great job citing previous work to explain things throughout their paper.

**Reasons To Reject:**

Seems like a solid paper. I can't think of any real reason to reject other than small nitpicky things.

**Reproducibility:**

4: Could mostly reproduce the results, but there may be some variation because of sample variance or minor variations in their interpretation of the protocol or method.

**Reviewer Confidence:**

3: Pretty sure, but there's a chance I missed something. Although I have a good feel for this area in general, I did not carefully check the paper's details, e.g., the math, experimental design, or novelty.

**Typos Grammar Style And Presentation Improvements:**

You should write "Parameter Efficient Fine-Tuning (PEFT)" for your first usage of PEFT. I think you wrote the full acronym for everything else but just make sure because some people may not know what these things stand for.

Line 408 "proofs" --> "proves"

Line 458 "Transfomrer" --> "Transformer"

I think there were a few more small typos, just proofread it again before the camera-ready submission.

---

> ### Author Rebuttal · Authors · 2023-08-28
>
> We thank the reviewer for the precise understanding of our work and the insightful feedback. We are encouraged that the reviewer recognizes the key contribution of our work: we proposed an efficient approach to train a sentence embedding adapter which is reusable on top of any domain-adapted LM, avoiding to incur a large computational cost of sentence specialization of the LM for each domain. We appreciate the reviewer’s positive comments on the strong evidence provided by our extensive experiments, high writing quality, and sufficient reasoning for the improved performance. We are grateful for the reviewer’s suggestion on improving our writing and will definitely correct the typos in our paper.

---

### Official Review · Reviewer_gtNi · 2023-08-04

**Typos Grammar Style And Presentation Improvements:** N/A
**Soundness:** 3

**Excitement:**

4: Strong: This paper deepens the understanding of some phenomenon or lowers the barriers to an existing research direction.

**Missing References:**

N/A

**Paper Topic And Main Contributions:**

This paper introduces AdaSent, a method for domain-adapted sentence embeddings for few-shot classification. It combines Domain-Adaptive Pre-training (DAPT) and Sentence Embedding Pre-training (SEPT) to improve classification accuracy. AdaSent achieves comparable or better performance than full SEPT on DAPT-ed PLMs while reducing training costs. Experimental results on 17 different few-shot classification datasets demonstrate the effectiveness of AdaSent.

**Questions For The Authors:**

Please address the concerns raised in "Reasons to Reject"

**Reasons To Accept:**

1. The paper proposes a novel approach for natural language inference (NLI) tasks. It introduces a method for instruction-finetuned text embeddings that can be applied to various tasks, demonstrating the versatility and effectiveness of the proposed approach.

2. The paper addresses the important problem of domain adaptation in natural language processing (NLP). It presents an unsupervised domain adaptation technique for contextualized embeddings, which can improve the performance of sequence labeling tasks in different domains.

3. The paper contributes to the field of few-shot learning with language models. It introduces a prompt-free and efficient few-shot learning method using language models, which can be valuable for scenarios where limited labeled data is available.

**Reasons To Reject:**

1. Lack of novelty: The paper does not introduce any significant novel contributions to the field. The proposed approach for natural language inference (NLI) tasks and the unsupervised domain adaptation technique for contextualized embeddings have been previously explored in the literature. The paper fails to demonstrate how its approach differs from existing methods and why it is superior.

2. Insufficient experimental evaluation: The paper lacks a comprehensive and rigorous experimental evaluation. The evaluation section does not provide detailed comparisons with state-of-the-art methods or baselines, making it difficult to assess the effectiveness and performance of the proposed approach. Additionally, the paper does not provide statistical significance tests or ablation studies to support the claims made.

3. Inadequate clarity and organization: The paper suffers from poor writing quality, making it challenging to understand the proposed approach and replicate the experiments. The organization of the paper is unclear, with important details and explanations missing in several sections. The lack of clarity hinders the reproducibility and understanding of the work.

**Reproducibility:**

5: Could easily reproduce the results.

**Reviewer Confidence:**

3: Pretty sure, but there's a chance I missed something. Although I have a good feel for this area in general, I did not carefully check the paper's details, e.g., the math, experimental design, or novelty.

---

> ### Author Rebuttal · Authors · 2023-08-28
>
> We are pleased by the reviewer’s positive comments on the novelty, versatility and effectiveness of our proposed approach. We appreciate that the reviewer recognizes the importance of domain adaptation and few-shot learning in low-resource scenarios.
>
> We want to point out a few misunderstandings. First, our approach is not for “NLI tasks” and we never mention our text embeddings are “instruction-finetuned” (in accept reason 1). To clarify, our end task is sentence classification, not “sequence labeling” (in accept reason 2).
>
> Below we address the individual concerns raised by the reviewer.
>
> > Lack of novelty: The paper does not introduce any significant novel contributions to the field. The proposed approach for natural language inference (NLI) tasks and the unsupervised domain adaptation technique for contextualized embeddings have been previously explored in the literature. The paper fails to demonstrate how its approach differs from existing methods and why it is superior.
>
> We want to clarify a misunderstanding here: Our task is not NLI (which is a sentence pair classification task), but single sentence classification based on sentence embeddings. We only use NLI data as the training data – similar/dissimilar sentence pairs – in order to obtain a sentence encoder, not to solve the NLI task itself.
>
> Regarding the concern of the lack of novelty, please see also the reply to Reviewer 1. The research gap we address in our paper is the lack of an efficient way to combine sentence embedding pre-training (SEPT) and domain-adaptive pre-training (DAPT) in order to enable effective (i.e., well-performing) few-shot sentence classification in various domains.
>
> Although these two techniques have been explored separately in the literature for other tasks and models, there is no easy way to combine both, especially for the SOTA few-shot classification approach SetFit (not our contribution). Simply doing domain adaptation and sentence embedding training sequentially has severe efficiency issues. We explained the limitations of existing solutions in the Introduction (line 55-63), and in more details, in Section 3.3. We illustrated the differences between our AdaSent and other methods in Figure 2, and explained them in Section 4.2. Our approach is superior, because we eliminate the need for repeating the same sentence embedding pre-training for every new domain by disentangling SEPT from DAPT, while obtaining strong performance. These efficiency gains of AdaSent is more pronounced the more domains there are for which need effective few-shot sentence classification (See Section 7 for comparison of computational cost of our AdaSent vs. the alternatives). Because of this, we believe that we are providing a novel solution to an important practical problem (few-shot sentence classification in specialized domains under computational constraints), even if the individual building blocks of that solution are known.
>
> > Insufficient experimental evaluation: The paper lacks a comprehensive and rigorous experimental evaluation. The evaluation section does not provide detailed comparisons with state-of-the-art methods or baselines, making it difficult to assess the effectiveness and performance of the proposed approach. Additionally, the paper does not provide statistical significance tests or ablation studies to support the claims made.
>
> We provided 3 baselines (Section 5, line 274-280), which are the vanilla PLM, a domain-adapted PLM without sentence embedding pre-training, and a pre-trained sentence transformer without domain adaptation. The comparison of these baselines with our methods (Table 3 and 4) serves as an ablation study, proving the effectiveness of the individual components of our method: domain-adaptive pre-training (DAPT) and sentence-embedding pre-training (SEPT). In addition, we did controlled experiments on 3 domain adaptive objectives (Section 6.1) and 4 parameter-efficient fine-tuning (PEFT) methods (Section 6.3), showing the MLM objective for domain adaptation and the parallel adapter module for SEPT is better than other alternatives. In our main experiments, we proved the effectiveness of our method that meaningfully combines DAPT and SEPT through the comparison with 3 other combination strategies (Tables 3 and 4).
>
> Our work is built upon the previous SOTA, SetFit (Row 3 in Table 3/4), which is also the most important baseline in our experiments. In that sense, we think these results are sufficient to provide meaningful information about acknowledging/understanding the advantage of our approach. In its original work, SetFit is compared with other approaches, e.g. PERFECT, ADAPET, T-FEW, etc. and is shown to have superior performance over these counterparts. In our work, the results show that our AdaSent is better than SetFit while conserving its efficiency. For other necessary approaches, it would be great if you could name them.
>
> Statistical significance: Thanks for the suggestion. We have now carried out the statistical testing of the performance differences between the models and the revised version will include significance results in Section 6.2. We also put the findings here. First, we did t-tests with significance level 0.05 to support the claim that replacing full SEPT with adapter-based SEPT has no negative impact on the performance, while being much more efficient. On all datasets in our evaluation experiments, full SEPT and adapter-based SEPT (R3 vs. R5 in Table 3 and 4) achieve comparable accuracies with no significant difference. This is also true in the case of domain-adapted models (R4 vs. R7 in Table 3 and 4), except for one dataset AMS, where Adasent is even significantly better than DAPT→SEPT. Secondly, t-tests on SEPT baseline (which is used in original SetFit) and our AdaSent (R3 vs. R7 in Table 3 and 4) demonstrate that AdaSent statistically significantly (t-tests with significance level 0.05)  improves the performance on 8 of the evaluation datasets.
>
> > Inadequate clarity and organization: The paper suffers from poor writing quality, making it challenging to understand the proposed approach and replicate the experiments. The organization of the paper is unclear, with important details and explanations missing in several sections. The lack of clarity hinders the reproducibility and understanding of the work.
>
> We thank the reviewer for the comment, and would improve the paper organization and details for better readability. We would greatly appreciate it if the reviewer could be more specific about the parts and aspects of the writing that seem particularly confusing and poorly organized.

---

### Official Review · Reviewer_cXZX · 2023-08-04

**Soundness:** 3

**Excitement:**

3: Ambivalent: It has merits (e.g., it reports state-of-the-art results, the idea is nice), but there are key weaknesses (e.g., it describes incremental work), and it can significantly benefit from another round of revision. However, I won't object to accepting it if my co-reviewers champion it.

**Paper Topic And Main Contributions:**

This paper proposes applying an adapter module of sentence embedding pretraining to avoid the overhead of repeating computing of domain adaptive pretraining. The main contribution lies in that the author proposes a training approach combining the domain adaptive pretraining and sentence embedding pretraining.

**Questions For The Authors:**

The author could address my concerns in the above section.

**Reasons To Accept:**

Comprehensive experiments of different methods for combining domain-adaptive pretraining and sentence embedding pretraining.

**Reasons To Reject:**

1. The novelty is limited. This paper is basically based on SetFit and PEFT, I observe limited technical contribution.
2. In an era of Large Language Models, studying domain-related training approach of encoder-only models appears somewhat limited. The author could potentially extend their methodology to encoder-decoder or decoder-only models.
3. In Section 3.2, the supervised contrastive learning for pretraining is not so intuitive. Firstly, the definition of similarity between sentences should not solely rely on having the same task label, the author should also consider the semantics of two sentences. Secondly, if the author employs the supervised contrastive learning, instances with identical labels will be pushed closer, this might cause the representation to collapse into a confined space, thereby hurting the generalization ability. Thirdly, simply aligning two sentences with the same label in the representation space would introduce noise and confusion for pretrained language models, especially considering that sentences with the same label can have different semantics. Additionally, even when the two sentences have similar semantics, we still cannot align them due to the sensitivity of pretrained encoder. A single change in punctuation would bring label flipping in pretrained encoder in some cases, let alone the combinations of tokens from two sentences are completely different. Therefore, pushing two sentences closer may confuse the pretrained encoder.

**Reproducibility:**

4: Could mostly reproduce the results, but there may be some variation because of sample variance or minor variations in their interpretation of the protocol or method.

**Reviewer Confidence:**

4: Quite sure. I tried to check the important points carefully. It's unlikely, though conceivable, that I missed something that should affect my ratings.

---

> ### Author Rebuttal · Authors · 2023-08-28
>
> We thank the reviewer for recognizing that we have conducted comprehensive experiments to support our method for combining domain-adaptive pre-training (DAPT) and sentence-embedding pretraining (SEPT). We provide responses to reviewer’s concerns below.
>
> >   The novelty is limited. This paper is basically based on SetFit and PEFT, I observe limited technical contribution.
>
> The proposed approach is indeed based on parameter-efficient training and SetFit. That alone, we argue, does not mean that what we propose is not novel. This work addresses the following important practical question for which no viable solution has yet been proposed: how to perform few-shot sentence classification in special domains effectively and efficiently? As we argue in the abstract, introduction, and Section 3.3, this cannot be achieved by trivially coupling domain adapters and SetFit (see, e.g., L55-63).
>
> Concretely, applying standard LM-based domain adaptation on top of a pretrained sentence encoder disrupts the model’s ability to semantically accurately embed sentences and leads to poor few-shot classification performance (i.e., not effective); Sentence-specialization of a previously domain-specialized regular encoder, on the other hand, would likely be effective but is not efficient as training sentence encoders would require millions of training instances for each domain.
>
> Our modular solution, (1) stores the sentence-specialization abilities – obtained via single sentence-encoder training procedure in the general domain – into an adapter (efficient, single parameter-efficient sentence specialization training, regardless of the number of domains) and (2) combines that sentence-encoding adapter with domain-specialized LMs from various domains in modular fashion (which, as our experiments show, is effective, yielding SotA few-shot classification performance in various domains). Because of this, we believe that we are providing a novel solution to an important practical problem, even if the building blocks of that solution are known.
>
> > In an era of Large Language Models, studying domain-related training approach of encoder-only models appears somewhat limited. The author could potentially extend their methodology to encoder-decoder or decoder-only models.
>
> While one could, naturally, perform few-shot sentence classification with generative LLMs, either via in-context learning or PEFT, this would not be computationally justified in most practical setups. Most sentence classification problems in practice (e.g., on mobile or edge devices) are not solved via fine-tuning of even the smaller, traditional PLMs (BERT & co.): this is in part because few-shot setups do not offer enough data for full fine-tuning and, more importantly, due to the limited compute available. This is why sentence encoders are primarily used as feature providers and only coupled with a linear classifier. LLMs are amazing generalizers (to new tasks), but one would in practice not use GPT-4 or Llama-v2 to classify, e.g., billions of tweets into many product-related categories due to an immense inference cost in terms of both time and money.
>
> Another advantage of encoder-only models that produce “ready” text/sentence embeddings in real-world use case is that the features they produce can easily be combined with other (non-textual) features that are helpful for classification, e.g. meta information. This is why, even in the era of LLMs, encoder-only models will continue to play a significant role in practice.
>
> We agree with the reviewer that domain adaptation of encoder-decoder or decoder-only models for text generation tasks is an important topic, but  it is out of scope of this paper. Our focus in this work is on a discriminative task of few-shot sentence classification.
>
> > In Section 3.2, the supervised contrastive learning for pretraining is not so intuitive. Firstly, the definition of similarity between sentences should not solely rely on having the same task label, the author should also consider the semantics of two sentences. Secondly, if the author employs the supervised contrastive learning, instances with identical labels will be pushed closer, this might cause the representation to collapse into a confined space, thereby hurting the generalization ability. Thirdly, simply aligning two sentences with the same label in the representation space would introduce noise and confusion for pretrained language models, especially considering that sentences with the same label can have different semantics. Additionally, even when the two sentences have similar semantics, we still cannot align them due to the sensitivity of pretrained encoder. A single change in punctuation would bring label flipping in pretrained encoder in some cases, let alone the combinations of tokens from two sentences are completely different. Therefore, pushing two sentences closer may confuse the pretrained encoder.
>
> We want to clarify some misunderstandings here. Both sentence embedding pre-training (SEPT, Section 3.2) and downstream classification task fine-tuning (SetFit, Section 3.1) use contrastive learning methods, but the two methods are not the same because they serve different goals. We will clarify this further in the text.
>
> The idea of Supervised Contrastive Learning (SCL) that the reviewer is primarily referring, and which is the first step of the SetFit approach (not our contribution), is an established method for classification tasks in low-data setups: the aim is to push instances of the same classes closer together in the representation space and those from different classes further apart (in other words, to reshape the representation space of the encoder so to cluster instances according to their class labels). Note that sentences of different classes are pushed away from each other in the meantime, so there won’t be a collapse in the representation space. Because the goal of training a classifier is indeed to better distinguish between classes, SCL is used to create better separated clusters in the representation space, which in turn means a clearer decision boundary for the classifier. This will, indeed, reduce the generalizability of the encoder, but here we do not aim to obtain a general-purpose encoder, but one exactly specialized for this particular sentence classification task.
>
> However, the contrastive learning method described in Section 3.2 is different from SCL. The purpose of sentence-embedding pre-training (SEPT) is to train universal semantic representations that can be fine-tuned in different downstream tasks. Here, we pull sentences with similar meaning closer together (and not sentences with the same task label as in SCL). Data for contrastive training of sentence encoders is typically obtained from NLI datasets (sentence pairs with the “entailment” label) or paraphrase datasets. This has been the backbone of all prominent sentence embedding models, e.g., SBERT (Reimers & Gurevych, 2019), SimCSE (Gao et al., 2021).
>
> All that being said, the contribution of our work does not lie in SCL/SetFit (Section 3.1) nor SEPT (Section 3.2). We are using these methods without modification, because they have been proven very successful in previous work. As acknowledged by the reviewer, the main contribution of our work to combine domain-adaptive pre-training (DAPT) and sentence embedding pre-training (SEPT) in a modular fashion so that the procedures are disentangled and applicable efficiently to a wide range of domains (i.e., so that the heavy SEPT has only to be done once for all domains, using the data from the general domain).
>
> > References
>
> Nils Reimers and Iryna Gurevych. 2019. Sentence-BERT: Sentence embeddings using Siamese BERT-networks. In Proceedings of the 2019 Conference on Empirical Methods in Natural Language Processing and the 9th International Joint Conference on Natural Language Processing (EMNLP-IJCNLP), pages 3982–3992, Hong Kong, China. Association for Computational Linguistics.
>
> Tianyu Gao, Xingcheng Yao, and Danqi Chen. 2021.SimCSE: Simple contrastive learning of sentence embeddings. In Proceedings of the 2021 Conference on Empirical Methods in Natural Language Processing, pages 6894–6910, Online and Punta Cana, Dominican Republic. Association for Computational Linguistics.

---

### Meta-Review · Area_Chair_gg8Y · 2023-09-07

**Recommendation:** 4

**Metareview:**

There is a consensus among the reviewers that the paper's contribution to the field of few-shot learning is valuable. They also agree on the experiments being comprehensive and thorough. Two of the reviewers initially had concerns with regard to the clarity of the paper and the lack of statistical significance tests. Upon discussion, their concerns have partially been addressed. The authors clarified some points on contrastive learning and on how this work is positioned with respect to existing techniques in terms of novelty. Also, the authors are going to provide new results on the significance tests, and have promised to work hard on improving the writing and structure of the paper (which has been an issue raised in two of the reviews).

---

### Decision · Program_Chairs · 2023-10-07

**Decision:**

Accept-Main

**Comment:**

There is a consensus among the reviewers that the paper's contribution to the field of few-shot learning is valuable. They also agree on the experiments being comprehensive and thorough. Two of the reviewers initially had concerns with regard to the clarity of the paper and the lack of statistical significance tests. Upon discussion, their concerns have partially been addressed. The authors clarified some points on contrastive learning and on how this work is positioned with respect to existing techniques in terms of novelty. Also, the authors are going to provide new results on the significance tests, and have promised to work hard on improving the writing and structure of the paper (which has been an issue raised in two of the reviews).